# Fault Tree Analysis and Failure Diagnosis of Marine Diesel Engine Turbocharger System

**Vlatko Knežević** [1,*]**, Josip Orović** [1] **, Ladislav Stazić** [2] **and Jelena Čulin** [1]

1   Maritime Department, University of Zadar, Mihovila Pavlinovića 1, 23000 Zadar, Croatia; jorovic@unizd.hr (J.O.); jculin@unizd.hr (J.Č.)
2   Faculty of Maritime Studies, University of Split, R. Boškovića 37, 21000 Split, Croatia; lstazic@pfst.hr
*   Correspondence: vknezevi1@unizd.hr

**Abstract:** The reliability of marine propulsion systems depends on the reliability of several sub-systems of a diesel engine. The scavenge air system is one of the crucial sub-systems of the marine engine with a turbocharger as an essential component. In this paper, the failures of a turbocharger are analyzed through the fault tree analysis (FTA) method to estimate the reliability of the system and to predict the cause of failures. The quantitative method is used for assessing the probability of faults occurring in the turbocharger system. The main failures of a scavenge air sub-system, such as air filter blockage, compressor fouling, turbine fouling (exhaust side), cooler tube blockage and cooler air side blockage, are simulated on a Wärtsilä-Transas engine simulator for a marine two-stroke diesel engine. The results obtained through the simulation can provide improvement in the maintenance plan, reliability of the propulsion system and optimization of turbocharger operation during exploitation time.

**Keywords:** reliability; fault tree analysis; failure diagnosis; diesel engine turbocharger; maintenance

## 1. Introduction

The reliability and safety of marine propulsion have a major role during the exploitation period which cannot be neglected. The safe operation of the vessel depends on the reliability of the main engine propulsion system. The reliability of any component or system is defined as the probability that a component or system will perform a required function for a given period of time when used under stated operating conditions [1]. The safety factor of a system is usually related to reliability and it can be defined as the avoidance of conditions that can cause injury, loss of life or damage to equipment and the surrounding environment [2]. Due to the complexity of the marine main engines and their sub-systems, it is difficult to predict when and how many failures will occur during a voyage.

For early detecting and avoiding unnecessary failures, the method of fault diagnosis is used. The main objectives of fault diagnosis are detection, isolation and fault analyses. The main tasks of fault diagnosis are to determine the type of faults, size, time of failure and localization of faults [3]. It is necessary during the exploitation of the ship to continuously monitor and record the technical conditions and parameters of all main and auxiliary equipment. Once the operating parameters have been assessed, the reliability and availability of any component can be estimated and measures for reducing the risk of failure can be considered. Furthermore, with a detailed failure diagnosis, the maintenance plan of any component of the system can be optimized and enhanced. An improved maintenance plan can reduce life-cycle costs such as the cost of preventive and corrective maintenance, cost of materials and energy and cost of spare parts transport and installation. Nowadays, the main challenge for turbocharger manufacturers is to increase efficiency in terms of fuel economy and environmental performance.

Failure diagnosis in this paper is focused on the turbocharger of a marine diesel two-stroke MAN 6S60MC-C engine. The purpose of this paper is to diagnose the most frequent symptoms of turbocharger faults using a deductive method fault tree analysis (FTA) and to simulate these failures on a Wärtsilä-Transas engine simulator to optimize operating conditions and improve the reliability of the turbocharger by avoiding undesirable events.

*Literature Overview*

Because turbochargers are the most important part of the scavenge system, they must have high reliability to ensure the reliability of the main engine, which is also concluded in research papers [4] and [5], where the reliability of the main engine subsystems is estimated, including turbocharger failure. It is concluded that turbocharger failure can have a major impact on the main engine operation and proper matching of the turbocharger and main engine is highly important. The matching method with an electric turbo compound for a two-stroke marine engine is proposed in article [6] and the method for the effective mistuning identification of marine turbocharger bladed discs is discussed in article [7]. These two methods can improve marine turbocharger efficiency but mostly in the manufacturing period.

Adamkiewicz [8] analyzed the relations between cause and effect of operation faults in a few turbocharger models with a method based on expert knowledge and operational diagnostic experience. Monieta [9] used a method with acceleration vibration signals for assessing the technical condition of turbochargers in three four-stroke engines. The research has shown that the diagnostic parameters of technical condition are more reliable with this method than the resource of the operating hours of the engine. In [10], marine propulsion system reliability is estimated using fault tree analysis. The failure probability of the entire ship propulsion system is hard to estimate due to the complexity of the system and each component has a specified life-cycle and maintenance interval. This method is suitable for the main engine subsystems or individual components of the engine.

The FTA method is used in a research study [11] for the risk assessment of the container terminal operations. The results have shown that human factors were the most common cause of accidents due to negligence in operating with equipment or vehicles. This method is also used in research paper [12] as a tool for modeling the marine main engine reliability.

The research paper [13] recommends reliability-centered maintenance (RCM) methodology to optimize the failure database of marine diesel engines. This methodology is useful to obtain an accurate and reliable database for predicting failures. However, this research is done on a four-stroke marine diesel generator with a power value of 1200 kW.

The analysis of failures during the early operation period of a ship is presented in article [14]. The observed marine engine was two-stroke, low speed and turbocharged, belonging to a bulk cargo ship and, moreover, failure analysis was conducted only during the first year of operation.

The various failures of marine engine operations are simulated in study [15] on a Konsberg Maritime engine simulator. The simulated incorrect engine operations were: worn and clogged injector nozzle, exhaust valve leakage, early injection timing. The importance of early-stage fault detection and efficiency management (planned maintenance) is emphasized.

In most cases, failure analysis and reviews are lacing simulation of faults during the operating conditions of a vessel. With this simulator-based methodology, it is possible to achieve more efficient operation of the engine, predict possible faults of the turbocharger system and develop enhanced maintenance intervals for the system.

## 2. Two-Stroke Marine Diesel Engine Turbocharger

The turbocharger in marine diesel engines is an essential element of the scavenge air system, which directly influences the power output, engine efficiency and emission of exhaust gases. The main components of the turbocharger are: turbine wheel (rotor with turbine blades), turbine nozzle ring,

steel shaft (turbine wheel on one end and compressor impeller assembled on the other), air compressor, silencer, diffuser, air filter and cooler.

In this paper, the focus is on the air filter, compressor impeller, turbine wheel and air cooler, because the engine room crew regularly inspects these components. Additionally, one of the important components of the turbocharger shaft that affects its reliability and durability are bearings. The key role of the bearing system is to control the radial and axial motion of the shaft and to reduce friction losses that have an impact on fuel efficiency. Furthermore, with new stringent emission regulations and demands, the lubricating oil viscosities become lower, so manufacturers must produce bearings that can maintain the stability of the rotor and avoid increased wear [16].

Leading manufacturers of turbochargers in the shipping industry are MAN Diesel & Turbo, ABB Turbocharging and Mitsubishi [17]. One of the new MAN TCR turbocharger models is shown in Figure 1. The new TCR turbocharger series has a wide range of applications, with engine power outputs from 390 to 7000 kW. The upcoming series is TCT which is specifically optimized for IMO Tier III engines, with a lighter design, superior charging efficiency and high air pressure.

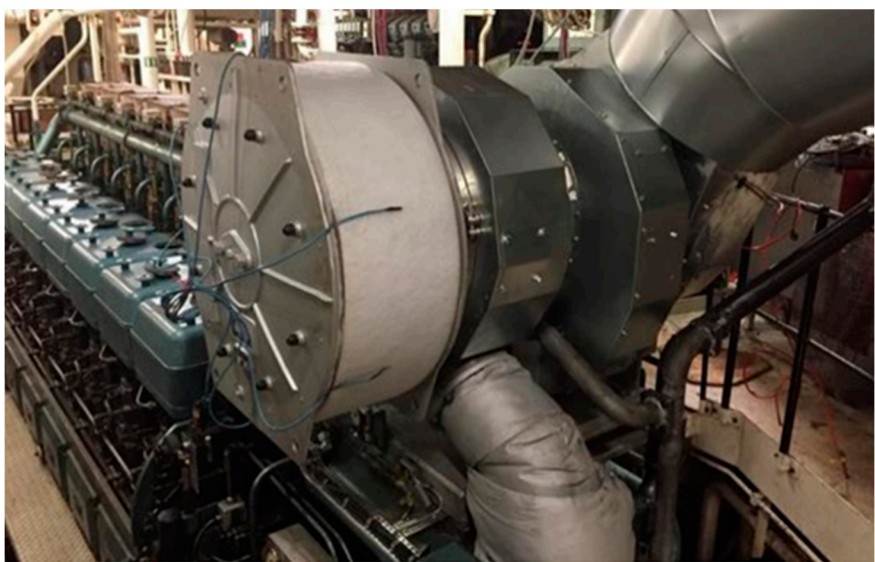

**Figure 1.** The turbocharger of two-stroke marine engine (MAN, TCR 18) [18].

Some turbochargers are designed with variable turbine area technology which enables the volume of charge air to be precisely matched to the quantity of fuel injected at all points regarding an engine's load and speed range [18]. Aside from mechanical innovations, there is research on using digital services that can provide reliable and simplified monitoring of turbocharger parameters to reduce maintenance costs.

The turbocharger of a simulated engine in this paper is a single stage type and it uses a charging system with constant pressure. Nowadays, manufacturers are trying to increase the energy efficiency of engines with two-stage turbochargers that use low- and high-pressure stages to deliver the charge air to cylinders at high pressure. Two-stage turbochargers can improve engine efficiency and reduce specific fuel oil consumption (SFOC), and these developments are important to satisfy new IMO NOx Tier regulations [19].

## 3. Fault Tree Analysis Method

The applied method for fault detection in this paper is FTA. The FTA method is a graphical model of the various combinations of faults that will result in the occurrence of the predefined undesired event [20]. With the FTA, the reliability of the marine propulsion system or any component of the sub-system can be estimated by calculating failure probability. The main purpose of FTA studies is

to develop comprehensive technology for early fault detection, system life prediction and enhanced maintenance intervals.

When creating a fault tree model, it is necessary to define the causal connections between events (failures) of the analyzed system, identify all possible faults that can cause the top event to happen and consider appropriate corrective measures. The reliability of the system (turbocharger) depends on the occurrence probability of undesired failures of its sub-units. In this case, it is the exhaust and air side of the turbine, air cooler and turbine shaft. Figure 2 shows a fault tree structure for the analyzed turbocharger.

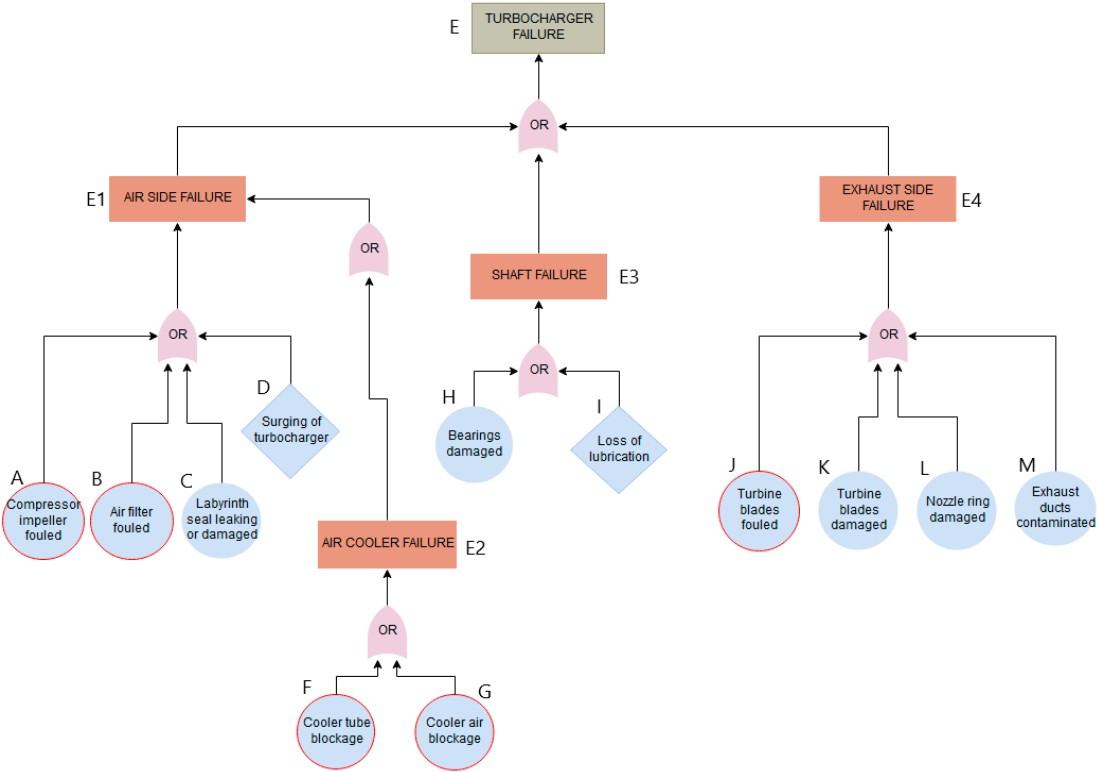

**Figure 2.** The fault tree of the turbocharger system.

The first step in the FTA is to define the top event, which is the most unwanted event in the system. Furthermore, it is important to determine all the events and conditions that leads to the top event. The fault events that lead to the top event are at the bottom and intermediate events are connected with logic gates. Logic gates represent the branches of the fault tree with their multiple inputs and just one output.

The primary events with their related symbols included in this fault tree model are:

- Basic events are faults events due to excessive operational stress resulting in the system element being out of operation [20]. These events do not need any further development and they are presented graphically with a circle (bottom events of fault tree). Basic events with a red outline are the ones that are simulated in this research.
- Intermediate events are faults that occur as a result of the combination through logic gates. They are symbolized by a rectangle and they pass through logic gates to the top event.
- Undeveloped events are specific faults that are not further developed, either because the event is of insufficient consequence or because information for the event is unavailable. They are graphically presented with a diamond.
- An OR gate is a logic gate that indicates that an output event occurs if one or more input events occur.

When the fault tree is constructed it can be used for assessing the probability of the basic events that are key parameters to determine the probability of the occurrence of the top event. For a better understanding of the interactive connections between basic and intermediate events, quantitative analysis is used and expressed with Boolean algebra. Using the probability theory, the fault tree model of the turbocharger can be expressed as:

$$P(E) = P(E1) + P(E3) + P(E4) \tag{1}$$

$$P(E1) = P(A) + P(B) + P(C) + P(D) + P(E2)$$

$$P(E2) = P(F) + P(G) - P(F \cap G)$$

$$P(E3) = P(H|I) = P(H)$$

$$P(E4) = P(J) + P(K) + P(L) + P(M) - P(J)P(K)P(L)P(M)$$

The occurrence of the top event P(E), is obtained as the sum of the fault probabilities of E1, E3 and E4. Basic events connected with an OR logic gate are considered as mutually exclusive (faults cannot occur at same time) or independent (occurrence of one event does not affect the occurrence of other events). The probability of event E1 can be calculated as the sum of mutually exclusive events (A, B, C and D) and the fault probability of event E2 that consists of two independent events. In the case of event E3, the probability of a fault is equal to the probability of basic event H, because event I is completely dependent on event H (loss of lubrication is unlikely to cause shaft failure without affecting bearing temperature). The occurrence of event E4 is defined as the probability fault sum of basic events that are all independent.

The probability of the occurrence of a fault event output from the "OR" gate can be calculated with the formula [21]:

$$P_{(y0)} = 1 - \prod_{i=1}^{k} \{1 - P(yi)\} \tag{2}$$

where:

$P(y_0)$: the probability of the occurrence of the OR gate output event
$k$: the number of input events in the OR gate
$P(yi)$: the probability of the input event in the OR gate. The input event is $yi = 1, 2, 3, \ldots, k$

## 4. Fault Simulation and Diagnosis

The faults (air filter blockage, compressor fouled, cooler tube and air side fouled, turbine fouled) are simulated using a Wärtsilä-Transas 5000 engine room simulator. The simulator provides a detailed copy of the vessel system and engine room models with interactive parameters and features for simulating exploitation conditions such as machinery faults or environment effects (wind, waves, hull fouling).

In this study, a propulsion plant of Tanker LCC (Aframax) with the main engine—MAN B&W 6S60 MC-C—is used (Figure 3). The main engine is two-stroke, low speed, reversible, crosshead type with six cylinders and constant pressure turbocharging. One turbocharger is fitted with equipment for washing the compressor and turbine side. Additionally, the engine is equipped with an air cooler for a fresh water cooling system. The type of fuel used for simulation is marine diesel oil (MDO) with a defined maximum sulfur content according to new IMO regulations. All mentioned faults are simulated while the engine is operating at 85% (nominal continuous rating) of maximum output with ambient air temperature set to 22 °C and humidity of 60%.

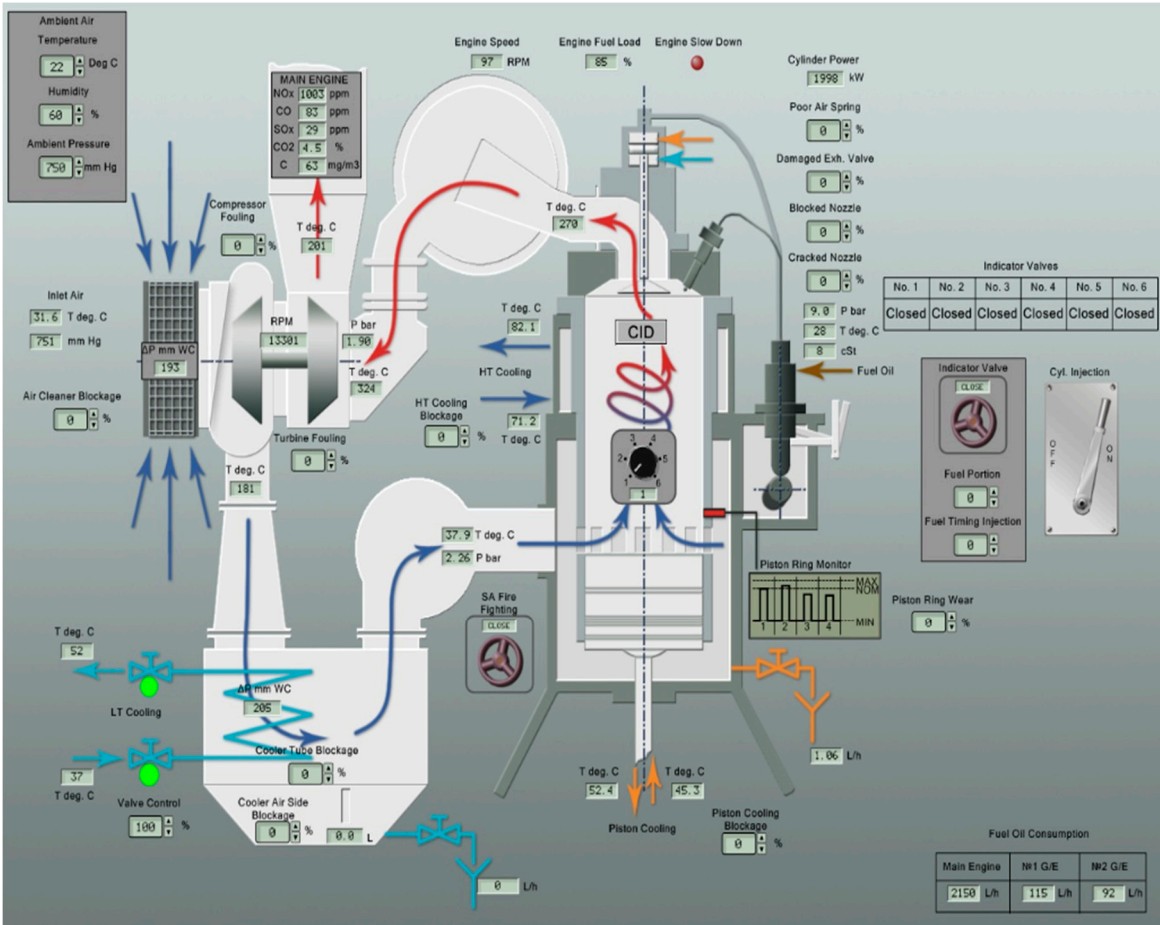

**Figure 3.** The interface of the main engine (cylinder) on Wärtsilä-Transas simulator [22].

*4.1. Air Filter Fault*

The blockage of the air filter is one of the most common faults associated with engine turbocharger systems. Fouling of the air filter and air flow ducts will significantly affect the quality of energy conversion and also it could cause an increase in fuel consumption. During operation, the air filter will eventually get contaminated and display the following inefficiencies [9]:

- Increase in the flow resistance
- Loss of filtering properties
- Loss of tightness

The amount of filter blockage can be set in the range of 0–100%, however, due to the automatic start of "slow down" operating mode (after 40% of filter blockage), faults of 10%, 20%, 30% and 40% of the fouled filter were simulated. The main engine parameters are presented in Table 1 for each fault.

**Table 1.** Air filter fouled.

| Main Engine Parameters | Fault—Air Filter Fouled | | | | |
|---|---|---|---|---|---|
| | 0% | 10% | 20% | 30% | 40% |
| Main engine inlet temperature (°C) | 31.60 | 31.70 | 31.80 | 32.40 | 32.70 |
| Turbocharger air filter pressure drop (mm WC) | 192.92 | 248.57 | 285.50 | 285.66 | 286.25 |
| Scavenge air inlet temperature (°C) | 181.12 | 173.49 | 165.57 | 141.51 | 124.46 |
| Scavenge air pressure drop (mm WC) | 205.27 | 200.66 | 194.62 | 153.53 | 110.47 |
| Scavenge air temperature manifold (°C) | 37.90 | 36.50 | 36.20 | 35.90 | 35.90 |
| Main engine scavenge air pressure (bar) | 2.26 | 1.99 | 1.77 | 1.41 | 1.31 |
| Average cylinder exhaust temperature (°C) | 258.49 | 274.44 | 292.48 | 353.55 | 484.33 |
| Exhaust gas outlet temperature (°C) | 200.82 | 219.88 | 241.47 | 315.62 | 456.35 |
| Turbocharger turbine inlet temperature (°C) | 323.49 | 336.48 | 351.48 | 405.67 | 530.47 |
| Turbocharger turbine inlet pressure (bar) | 1.90 | 1.70 | 1.50 | 1.20 | 0.91 |
| Turbocharger rpm (r/min) | 13,296 | 13,240 | 13,442 | 12,537 | 11,478 |
| Main engine rpm (r/min) | 96.60 | 96.60 | 96.60 | 96.60 | 96.50 |
| Main engine fuel load (%) | 84.48 | 84.48 | 84.48 | 84.51 | 85.55 |

The increase in filter fouling percentage results in less air supplied on the air side of the turbocharger, which is shown by these indicators in Table 1:

- slightly increased main engine inlet temperature
- increased air filter pressure drop
- reduced scavenge air inlet temperature
- reduced turbocharger scavenge air pressure drop
- reduced scavenge air temperature and pressure
- increased exhaust gas temperature on all cylinders
- increased turbocharger turbine inlet temperature
- reduced turbine inlet pressure
- reduced turbocharger rpm
- slightly increased main engine fuel load

A fault of the air filter could drastically affect the operation of the main engine if the amount (percentage) of fouling increases. Since deposits on the air filter increase, the pressure drop on the air filter is also increased, which results in ineffective compressor operation. The compressor supplies less fresh air into the cylinders due to the fouled filter, thus the main engine scavenge air pressure is reduced (2.26 bar to 1.31 bar) and the scavenge air temperature (181.12 °C to 124.46 °C) before the air cooler is also reduced. Furthermore, this fault has an impact on the increase in exhaust temperatures in the cylinders and exhaust temperature at the turbine inlet. Insufficient air supply also reduces pressure at the turbine inlet (1.90 bar to 0.90 bar), therefore, turbocharger revolutions (rpm) are reduced. The impact of an air filter fault on the main engine combustion process is explained in Section 5.1.

### 4.2. Air Cooler Faults

Turbochargers increase the temperature of the air in intake manifold, so it is important to reduce these excessive temperatures to achieve an efficient combustion process and lower exhaust emissions. For this purpose, engines are equipped with a scavenge air cooler which is usually constructed of bronze alloy tubes for cooling water circulation and aluminum fins for necessary air flow [23]. Some of the air coolers are cooled by sea water, but in this case, a low-temperature fresh water circuit is used.

The loss of the cooling efficiency of the air cooler is related to insufficient air flow (blockage of cooler air side) and ineffective cooling (cooler tube blockage). These two main faults are simulated with the amount of fouling from 0–50%. The main engine monitored parameters during cooler tube blockage are presented in Table 2.

**Table 2.** Cooler tube blockage.

| Main Engine Parameters | Fault—Cooler Tube Blockage | | | | | |
|---|---|---|---|---|---|---|
| | **0%** | **10%** | **20%** | **30%** | **40%** | **50%** |
| Main engine inlet temperature (°C) | 31.60 | 31.50 | 31.50 | 31.50 | 31.40 | 31.40 |
| Turbocharger air filter pressure drop (mm WC) | 192.92 | 197.50 | 203.51 | 211.51 | 220.53 | 232.52 |
| Scavenge air inlet temperature (°C) | 181.12 | 183.54 | 188.52 | 194.50 | 201.51 | 210.50 |
| Scavenge air pressure drop (mm WC) | 205.27 | 206.50 | 207.53 | 210.53 | 212.53 | 216.51 |
| Scavenge air temperature manifold (°C) | 37.90 | 47.90 | 60.00 | 74.90 | 92.90 | 115.50 |
| Main engine scavenge air pressure (bar) | 2.26 | 2.31 | 2.38 | 2.47 | 2.58 | 2.71 |
| Average cylinder exhaust temperature (°C) | 258.49 | 264.51 | 272.51 | 284.50 | 298.51 | 315.53 |
| Exhaust gas outlet temperature (°C) | 200.82 | 204.51 | 209.52 | 217.51 | 226.51 | 237.50 |
| Turbocharger turbine inlet temperature (°C) | 323.49 | 329.53 | 338.53 | 351.50 | 365.53 | 384.50 |
| Turbocharger turbine inlet pressure (bar) | 1.90 | 2.00 | 2.00 | 2.10 | 2.20 | 2.30 |
| Turbocharger rpm (r/min) | 13,296 | 13,407 | 13,564 | 13,785 | 14,070 | 14,432 |
| Main engine rpm (r/min) | 96.60 | 96.60 | 96.60 | 96.60 | 96.60 | 96.60 |
| Main engine fuel load (%) | 84.48 | 84.48 | 84.48 | 84.48 | 84.50 | 84.50 |

A blockage of cooler tubes will affect the cooling water flow in tubes which will result in a loss of cooling efficiency, as well as these unwanted indications:

- increased air filter pressure drop
- increased scavenge air inlet temperature
- increased turbocharger scavenge air pressure drop
- increased scavenge air temperature manifold
- increased average exhaust gas temperature on all cylinders and exhaust outlet temperature
- increased turbocharger turbine inlet temperature
- increased turbocharger rpm

With reduced cooling efficiency, temperatures of exhaust gases and scavenge air increase, especially the temperature of charge air (temperature manifold) before entering the cylinders. The reasons and effects of fouled cooler tubes are discussed in Section 5.5.

In Table 3, the main engine parameters for a cooler air side fault are presented. Fouling of the air side of the cooler will reduce the amount and quality of combustion air entering the cylinders, leading to these indications:

- slightly increased main engine inlet temperature
- reduced air filter pressure drop
- reduced scavenge air inlet temperature
- increased turbocharger scavenge air pressure drop
- reduced main engine scavenge air inlet pressure
- increased average exhaust gas temperature on all cylinders and exhaust outlet temperatures
- increased turbine inlet temperature
- reduced turbine inlet pressure
- reduced turbocharger rpm
- slightly increased main engine fuel load



**Table 3.** Cooler air side blockage.

| Main Engine Parameters | Fault—Cooler Air Side Blockage | | | | |
|---|---|---|---|---|---|
| | 0% | 10% | 20% | 30% | 40% |
| Main engine inlet temperature (°C) | 31.60 | 31.70 | 31.90 | 32.30 | 32.90 |
| Turbocharger air filter pressure drop (mm WC) | 192.92 | 166.47 | 133.48 | 99.35 | 71.48 |
| Scavenge air inlet temperature (°C) | 181.12 | 174.49 | 165.49 | 154.40 | 144.47 |
| Scavenge air pressure drop (mm WC) | 205.27 | 215.59 | 221.45 | 223.57 | 236.26 |
| Scavenge air temperature manifold (°C) | 37.90 | 36.60 | 35.90 | 35.90 | 35.90 |
| Main engine scavenge air pressure (bar) | 2.26 | 2.00 | 1.68 | 1.34 | 1.04 |
| Average cylinder exhaust temperature (°C) | 258.49 | 274.47 | 300.52 | 347.63 | 411.51 |
| Exhaust gas outlet temperature (°C) | 200.82 | 219.45 | 250.58 | 302.51 | 370.52 |
| Turbocharger turbine inlet temperature (°C) | 323.49 | 336.45 | 359.27 | 400.65 | 459.50 |
| Turbocharger turbine inlet pressure (bar) | 1.90 | 1.70 | 1.40 | 1.10 | 0.80 |
| Turbocharger rpm (r/min) | 13,296 | 12,688 | 11,841 | 10,840 | 9793 |
| Main engine rpm (r/min) | 96.60 | 96.60 | 96.60 | 96.60 | 96.60 |
| Main engine fuel load (%) | 84.48 | 84.48 | 84.48 | 84.51 | 85.53 |

The amount of fouling is set to a maximum of 40%, due to the automatic start of "slow down" mode when exhaust temperatures exceed their set limit point. Moreover, less air supplied to the cylinders reduces the pressure of the scavenge air before entering the cylinders and turbine inlet pressure.

*4.3. Compressor Wheel Fault*

The purpose of the turbocharger compressor is to draw air from the engine room in an axial direction and then expel it in a radial direction with high velocity. Three essential components of the compressor that ensure high performance are: compressor wheel, casing and diffuser.

Maintenance of the compressor side is highly important to avoid fouling of the compressor blades which can lead to excessive air flow resistance. The simulation of the fouled compressor wheel is shown in Table 4. The amount of fouling is set to 25%, 50%, 75% and 90%, unlike faults of the air filter and cooler where operating mode "slow down" automatically starts after a certain percentage of fouling.

**Table 4.** Compressor wheel fouled.

| Main Engine Parameters | Fault—Compressor Wheel Fouled | | | | |
|---|---|---|---|---|---|
| | 0% | 25% | 50% | 75% | 90% |
| Main engine inlet temperature (°C) | 31.60 | 31.60 | 31.60 | 31.60 | 31.70 |
| Turbocharger air filter pressure drop (mm WC) | 192.92 | 189.38 | 185.52 | 183.54 | 182.50 |
| Scavenge air inlet temperature (°C) | 181.12 | 199.54 | 225.50 | 262.50 | 292.54 |
| Scavenge air pressure drop (mm WC) | 205.27 | 223.16 | 247.51 | 284.55 | 317.51 |
| Scavenge air temperature manifold (°C) | 37.90 | 37.20 | 38.50 | 40.00 | 41.30 |
| Main engine scavenge air pressure (bar) | 2.26 | 2.22 | 2.19 | 2.17 | 2.16 |
| Average cylinder exhaust temperature (°C) | 258.49 | 258.60 | 261.50 | 264.50 | 267.52 |
| Exhaust gas outlet temperature (°C) | 200.82 | 201.52 | 204.53 | 207.50 | 210.50 |
| Turbocharger turbine inlet temperature (°C) | 323.49 | 323.58 | 325.51 | 328.50 | 331.51 |
| Turbocharger turbine inlet pressure (bar) | 1.90 | 1.90 | 1.90 | 1.90 | 1.90 |
| Turbocharger rpm (r/min) | 13,296 | 13,240 | 13,152 | 13,103 | 13,084 |
| Main engine rpm (r/min) | 96.60 | 96.60 | 96.60 | 96.60 | 96.60 |
| Main engine fuel load (%) | 84.48 | 84.48 | 84.48 | 84.48 | 84.50 |

Table 4 presents indications which will help to establish these shortcomings of the compressor wheel/impeller:

- reduced air filter pressure drop
- increased scavenge air inlet temperature
- increased turbocharger scavenge air pressure drop
- increased scavenge air temperature manifold
- slightly reduced main engine scavenge air pressure
- increased average cylinder temperatures and outlet exhaust temperature
- increased turbocharger turbine inlet temperature
- reduced turbocharger rpm

While the amount of deposits on the compressor wheel is increasing it affects compressor efficiency and main engine parameters. With the reduced efficiency of the compressor, the scavenge air pressure is insufficient and the most affected parameter is the temperature of the scavenge air inlet (charge air temperature after the compressor), which increases significantly (181.12 °C to 292.54 °C). The increase in the fouling percentage on the compressor blades also affects main engine performance in terms of increased exhaust temperatures in the cylinders and at the turbine inlet. Furthermore, with an excessive charge air temperature, the air cooler cannot efficiently reduce this temperature, so the quality of air entering the cylinders is inadequate for proper combustion processes. The simulated scenario with a high percentage (90%) of fouling could even lead to severe damage to the compressor impeller. Compressor maintenance and optimization are discussed in Section 5.2.

### 4.4. Turbine Blades Fault

The turbocharger turbine side, which consists of a turbine casing and turbine wheel, is a crucial part for converting exhaust gas energy into mechanical energy (shaft power) to drive the compressor. Because the high-velocity and high-temperature exhaust gas is directed onto the turbine blades, without preventive maintenance, the exhaust side of the turbocharger can easily get contaminated with carbon deposits and soot from the combustion process. Fouled turbine blades cause an increase in the exhaust gas flow resistance, which leads to a reduction in turbine efficiency and an increase in the specific fuel consumption. The main engine parameters during fouled turbine wheel simulation are presented in Table 5.

**Table 5.** Turbine wheel fouled.

| Main Engine Parameters | Fault—Turbine Wheel Fouled | | | | |
|---|---|---|---|---|---|
| | 0% | 25% | 50% | 75% | 90% |
| Main engine inlet temperature (°C) | 31.60 | 31.60 | 31.90 | 32.20 | 32.50 |
| Turbocharger air filter pressure drop (mm WC) | 192.92 | 171.68 | 144.50 | 114.47 | 93.57 |
| Scavenge air inlet temperature (°C) | 181.12 | 164.49 | 146.61 | 129.50 | 118.55 |
| Scavenge air pressure drop (mm WC) | 205.27 | 201.67 | 195.61 | 184.50 | 172.47 |
| Scavenge air temperature manifold (°C) | 37.90 | 36.30 | 35.90 | 35.90 | 35.90 |
| Main engine scavenge air pressure (bar) | 2.26 | 1.98 | 1.66 | 1.34 | 1.13 |
| Average cylinder exhaust temperature (°C) | 258.49 | 269.49 | 290.48 | 324.50 | 358.50 |
| Exhaust gas outlet temperature (°C) | 200.82 | 223.49 | 255.53 | 300.51 | 340.50 |
| Turbocharger turbine inlet temperature (°C) | 323.49 | 332.46 | 349.52 | 379.51 | 410.39 |
| Turbocharger turbine inlet pressure (bar) | 1.90 | 1.70 | 1.40 | 1.10 | 0.90 |
| Turbocharger rpm (r/min) | 13,296 | 12,370 | 11,989 | 11,087 | 10,327 |
| Main engine rpm (r/min) | 96.60 | 96.60 | 96.60 | 96.60 | 96.60 |
| Main engine fuel load (%) | 84.48 | 84.48 | 84.48 | 84.51 | 84.51 |

Simulated faults in the range of 25–90% reduce turbine output capacity and result in these important indications:

- reduced air filter pressure drop
- reduced turbocharger scavenge air pressure drop
- reduced scavenge air inlet temperature
- reduced main engine scavenge air pressure
- increased turbocharger turbine inlet temperature
- increased average exhaust gas temperature on all cylinders
- increased exhaust gas outlet temperature
- reduced turbocharger rpm

The fault (fouling) of turbine blades mostly depends on the quality of the fuel used and the combustion process in the cylinders. Incomplete fuel burning causes layers of deposits on the turbine blades which result in excessive exhaust gas temperatures. An increased amount of fouling percentage on the turbine blades reduces the pressure of exhaust gases at the turbine inlet stage (1.90 bar to 0.90 bar), thus the turbine does not have the necessary output power to provide charge air with constant pressure.

## 5. Results and Discussion

### 5.1. Turbocharger Air Side Results

The required amount and quality of intake air for proper combustion processes depends on the efficient and optimized operating conditions of the compressor wheel and air filter. The fouling of the air filter and compressor wheel are the two most common faults on the air intake side during engine operation and they could easily be detected by monitoring engine parameters. These changes in main engine parameters are presented in the previous section, however, the results of faults can also be shown on the cylinder indicator diagram. Indicator diagrams are used to assess the performance of each cylinder to detect any differences in the combustion process during the voyage.

The results of air filter fault simulation are presented with an indicator diagram (cylinder pressure/crank angle) in Figure 4. Cylinder indicator diagrams (Figures 4–7) are obtained using the simulator's built-in option for recording the pressure in each engine cylinder and an analyzing option for a comparison of recorded indicator diagrams according to the fault percentage. In the indicator diagram, the horizontal axis represents the cylinder crankshaft angle and the vertical axis is pressure in the cylinder.

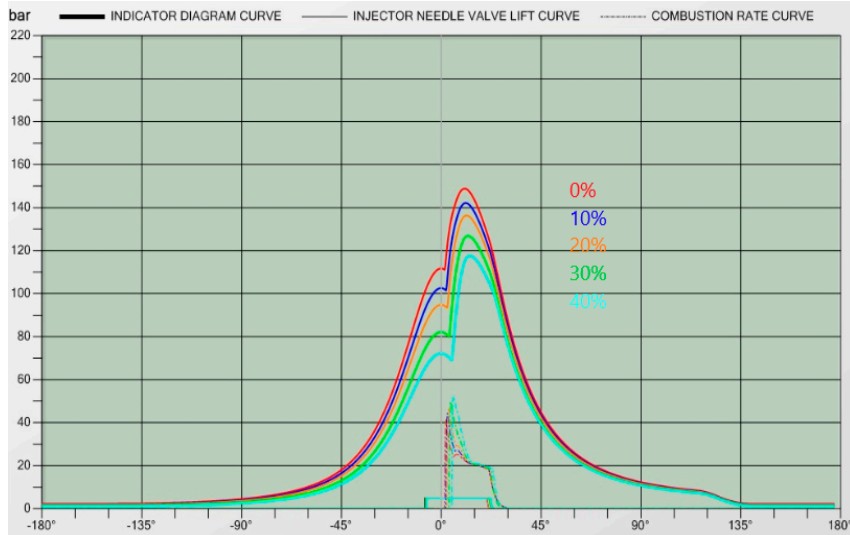

**Figure 4.** Indicator diagram during air filter faults.

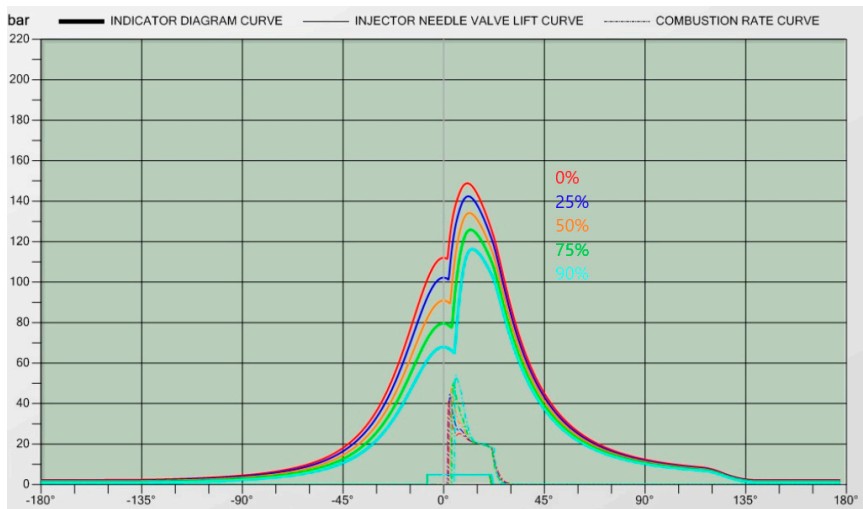

**Figure 5.** Indicator diagram for turbine blade faults.

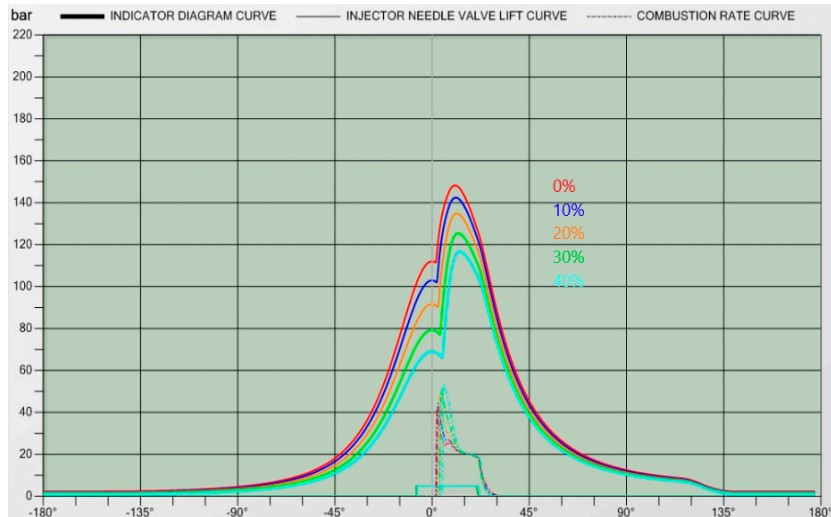

**Figure 6.** Indicator diagram of the fouled cooler—air side.

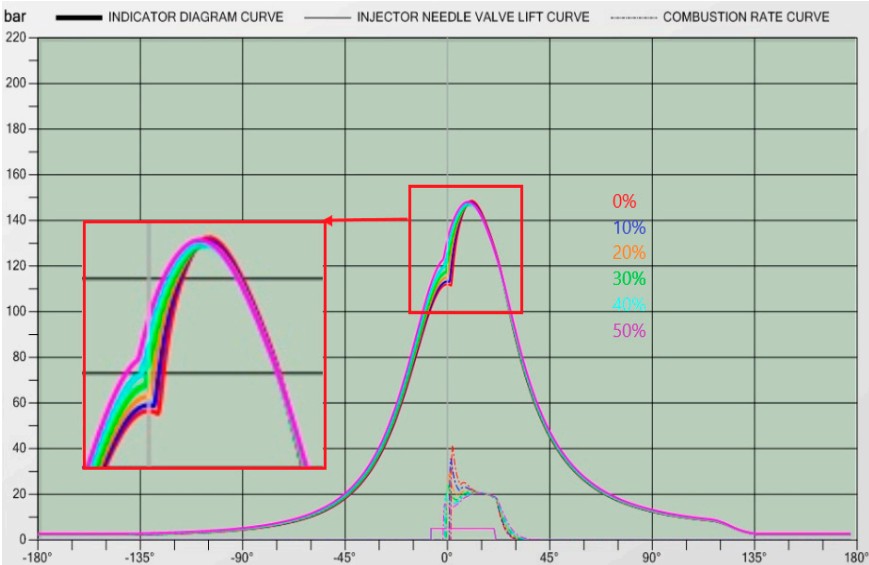

**Figure 7.** Indicator diagram of blocked cooler tubes.

As the contamination of the air filter increases, it affects the combustion pressures and injection timing crank angles. Each fault in the diagram is presented with different colors according to the fouling percentage, while the red line (0%) is a simulation in normal operating conditions. Analyzing the indicator diagram, the main differences during the combustion process in the cylinder are:

- reduced maximum combustion pressure from 150 bar to 119 bar
- reduced compression pressure from 112 bar to 73 bar
- increased angle of combustion start (1.5° to 4.7°), resulting in a late combustion process
- reduced (late) timing angle of fuel injection (−7.4° to −6.5°)

The fouling of the air filter and compressor impeller is usually from fuel and oil vapors, cargo residues and dust from the engine room. These are the reasons for increasing the flow resistance on the air side of the turbocharger, which eventually reduces the amount of air supplied for the combustion process and overall efficiency of the turbocharger.

With insufficient air supplied to the cylinders, the combustion process is improper, resulting in black smoke from the exhaust and increased fuel consumption. Because the amount of air for each cylinder is reduced, increased fuel injection causes an increase in exhaust gas temperatures and, eventually, activation of the alarm for high exhaust gas temperature and main engine slow down operating mode.

## 5.2. Optimization and Maintenance of Air Filter and Compressor Wheel

The adequate maintenance of each turbocharger component is highly important, not only for preventing failures but also to extend the durability and reliability of the system. Failures due to improper maintenance are usually caused by human errors and they are hard to predict or avoid. The assessment of human reliability during turbocharger maintenance procedures is presented in [24] and an evaluation of human error probabilities in [25], with emphasis on contributing factors for making errors such as a high level of noise and vibration, weather conditions, level of ship motion and stress. The probability of human error during the cleaning of the air filter is very low because it is an easy task to perform and an old filter with a silencer can be replaced or washed onboard.

Turbocharger manufacturers recommend maintenance and inspection intervals for each component according to engine operating hours. For an air filter on a two-stroke engine, the maintenance interval (clean air filter/depending on condition) is set to every 250 h [18]. Controlling the contamination of the air filter should be a crucial part of the inspection interval. The results of an air filter fault have shown that it drastically affects the main engine and turbocharger parameters, so the recommendation is to enhance the maintenance interval by checking the fouling of filter or unusual vibrations more often and to compare scavenge air pressure differences on the manometer.

The contamination of the compressor wheel and air intake casing is usually caused by lubricating oil vapors (entering through the labyrinth seal) or particles of fuel contained in the air. Moreover, due to the high speed of the turbocharger, foreign objects, parts of the equipment or even small particles in air flow ducts could lead to serious damage to the compressor wheel. A compressor wheel with damaged impeller blades affects the aerodynamics of the air flow and it results in insufficient scavenge air pressure and a reduction of compressor efficiency.

Another problem that can also affect compressor performance is surging (specific fault/event on fault tree). Surging occurs when the air pressure charged by the compressor is higher than the pressure inside the compressor and it creates a reverse air flow towards the inlet of the compressor. This deviation of the pressure is hard to predict and it could be due to sudden changes in the main engine load or imbalanced or damaged blades. In [26], a detailed measurement of engine performance during compressor surge is analyzed, with the conclusion that in marine engines with intake manifold of large volumes, it is hard to prevent the surging effect.

To optimize compressor working performance, it is necessary to evaluate the compressor efficiency. Compressor efficiency is defined as the ratio of the work a compressor performs under insulated conditions to that of an compressor under actual conditions, and it is expressed with Equation (3) [27]:

$$\eta_{is,c} = \frac{\Delta h_{is,c}}{\Delta h_c} = \frac{C_{p,c} \cdot (T_{2,is} - T_1)}{C_{p,c} \cdot (T_2 - T_1)} \tag{3}$$

The fouling of the compressor wheel does not affect the combustion pressures or timing and duration of fuel injection as an air filter fault. However, it significantly increases the compressor outlet temperature (in Table 3: scavenge air inlet temperature). This temperature is indicated as $T_2$ in Equation (3), and when it increases, the compressor efficiency is reduced.

To prevent this from happening, period maintenance is required at scheduled intervals. The most common methods for cleaning the compressor or turbine blades are wet and dry cleaning. The dry cleaning method is carried out with compressed air blown at the compressor wheel. However, this method is not recommended for heavier deposits. Manufacturers recommend wet cleaning (fresh water) for compressor blades while the engine is at full load. Usually, compressor maintenance is neglected or postponed until dry-docking rather than performing maintenance during operating hours. Sometimes neglected washing routines can lead to an increase in dirt deposits on both the compressor and turbine blades and this could cause an imbalance of the rotor or even bearing damage. The recommendation is to continuously monitor scavenge air temperature and perform the cleaning at intervals adjusted to the amount of contamination.

### 5.3. Turbocharger Exhaust Side Results

The exhaust side of the turbocharger consists of the gas inlet and outlet casing, nozzle ring and turbine rotor with blades for converting exhaust gas kinetic energy into mechanical energy. Turbine blades are directly exposed to a high temperature of exhaust gases, therefore, their condition depends on the quality of the combustion process and fuel used. Due to improper combustion, unburnt carbon and soot particles in exhaust gas can cause fouling and damage to the nozzle ring and blades. Moreover, severe damage to turbine blades can be caused by pieces of broken piston rings or valves.

The faults (fouling) of turbine blades also cause differences in the combustion process and they are presented in Figure 5.

The fouling of turbine blades leads to a reduction of turbine output capacity and overall efficiency. The main changes in the combustion process from the cylinder diagram are:

- reduced mean effective pressure (20.1 to 19.3 bar)
- reduced maximum combustion pressure (150 to 117 bar)
- reduced compression pressure from (112 to 68 bar)
- increased angle of combustion start (1.5° to 4.6°), resulting in late combustion process

In the case of a high fouling level, exhaust gas temperatures (at turbine inlet) are increased and the turbocharger consequently supplies less charged air in the cylinders, so the combustion process starts later and lasts longer.

### 5.4. Maintenance and Optimization of the Turbocharger Exhaust Side

To ensure the durability and efficiency of the turbocharger, maintenance of the turbine rotor, nozzle ring and blades at regular intervals is highly important. For cleaning the deposits on turbine blades, two methods that can be used without the need to stop the engine are the wet and dry cleaning methods. Usually, for turbine cleaning, the wet method is applied due to better cleaning effects and longer maintenance intervals [18].

For two-stroke engines, a maintenance interval of 150 operating hours is recommended [6], however, it should be adapted according to the quality of the fuel used. The fresh water for cleaning

should be without any chemical additives and sprayed into the exhaust gas casing before the turbine at a pressure of 2 or 3 bar. In a scenario with high exhaust temperatures during wet cleaning, it is necessary to reduce engine load to avoid the thermal stress of turbine materials. The advantage of dry cleaning is that it can be carried out during operation at full load, however, heavier deposits are harder to remove and maintenance intervals are shortened.

The efficiency of the turbine depends on the energy in exhaust gases which is converted into turbine power output for the intake of air mass flow from the compressor side.

Turbine power is expressed with Equation (4) [27]:

$$P_t = \dot{m}_t \cdot C_{p,t} \cdot (T_3 - T_4) \tag{4}$$

In Equation (4), temperature $T_3$ represents the turbine inlet temperature and $T_4$ is exhaust outlet temperature. The results of the turbine fault have shown that both temperatures simultaneously increase proportionally to the amount of fouling. While these temperatures increase, the turbine power is reduced due to lower temperature difference ($\Delta T$) and turbine gas flow rate ($\dot{m}_t$). With insufficient turbine outlet power, turbocharger revolutions (rpm) are also reduced and consequently less fresh air is supplied to the cylinders. Furthermore, less air has a negative effect on the scavenge air pressure (reduced from 2.26 to 1.13 bar) and turbine inlet pressure (reduced from 1.90 to 0.9 bar).

The presented results of turbine fouling indicate the importance of preventive maintenance. The high amount of deposits in the exhaust side of the turbocharger will eventually lead to damage to the turbine. Moreover, when the turbine wheel is damaged, it is imbalanced and it could cause serious issues for the shaft and bearings. Corrective measures for repairing turbocharger damage are not an easy task in large two-stroke marine engines during voyage. The reasons are usually related to missing spare parts or insufficient crew maintenance knowledge and experience. To avoid failure of the exhaust side, it is recommended to adjust the maintenance interval (according to contamination level) and to regularly observe exhaust gas temperatures (before and after the turbine).

### 5.5. Air Cooler Results

Turbocharger efficiency also depends on the operating condition of the scavenge air cooler. The results of two simulated faults (fouling of air side and tubes) have shown that it could significantly affect the combustion process and these faults should not be neglected. The effects of fouling include loss of heat transfer (tube blockage) and a pressure drop decrease (air side fouled).

Fouling of the air side occurs due to dust and atmospheric particulates contained in the air and the fins of the cooler have a role as a filter where particulates can deposit. As the layers of deposits increase, less air is supplied to the cylinders, leading to reduced inlet air pressure (from 2.26 bar to 1.04 bar) and turbine inlet pressure (from 1.90 bar to 0.80 bar). With reduced turbine inlet pressure, the turbocharger does not have enough output capacity for the compressor side, so the air filter pressure drop is also reduced (from 192 to 71 mm WC). Therefore, less supplied air will increase the exhaust gas temperatures from the cylinders and fuel consumption. Differences in the combustion process for this fault are presented in Figure 6.

The results shown in the indicator diagram (Figure 6) are similar to the results of the air filter fault due to insufficient charged air flow. The symptoms of this fault in the combustion process are:

- reduced mean effective pressure (20.3 to 19.5 bar)
- reduced maximum combustion pressure (150 to 118 bar)
- reduced compression pressure from (112 to 70 bar)
- increased angle of combustion start (1.5° to 4.6°)

Regular maintenance for the air cooler involves the injection of cleaning additives with water. This mist of water and solvents is necessary for cleaning the air fin deposits which can reduce air flow. The washing down process is highly important to ensure all the contamination is flushed out and to improve scavenging efficiency and heat transfer.

The second simulated fault of the air cooler (tube blockage) also has a negative effect on the combustion process. The blockage of water flow through the tubes is related to the inadequate treatment of cooling water, which has corrosive and sediment-laden properties. When cooler tubes are blocked, the temperatures of scavenge air and exhaust gases are increased. The temperature of scavenge air during normal operating conditions is 37.90 °C, while during 40% tube blockage, it is increased to 92.90 °C (Table 2). Moreover, the combustion process in the cylinders is inefficient, therefore exhaust temperatures and turbine inlet pressure are increased from 1.90 to 2.20 bar. With higher pressure at the turbine inlet and excessive temperatures, turbocharger shaft revolutions are increased, which could lead to turbocharger overspeeding. The results of the combustion process in the indicator diagram are presented in Figure 7.

A high temperature of scavenge air in the combustion process causes these changes in the indicator diagram:

- increased compression pressure from (112 to 124 bar)
- reduced angle of combustion start (1.5° to −2.0°), resulting in the early start of the combustion process

To avoid mechanical damage to the fins and tubes of the air cooler, it is recommended to detect the level of fouling in the early stage. The easiest way to detect this is to measure and control the scavenge air temperature and pressure difference. Sometimes maintenance of the air cooler is neglected during voyage and it is postponed until major overhaul. The efficiency and reliability of the air cooler depends on regular maintenance intervals and the control of fouling layers in tubes and fins, especially in large two-stroke marine engines with an enormous mass of air flow charged into cylinders.

## 6. Conclusions

The reliability of the marine turbocharger system is crucial to ensure efficient performance during the exploitation period. High reliability of the turbocharger system depends on preventing failures by using a fault diagnosis method, engine performance evaluation and appropriate maintenance intervals. Usually, fault diagnosis during engine operation depends on the engine crew's experience, which can lead to false conclusions and improper corrective actions.

Before analyzing turbocharger faults, it is necessary to evaluate relations between the cause and symptoms of all the possible faults that can occur during operation. In this paper, the most common faults in each turbocharger component are simulated and analyzed. The main conclusions of this research are:

- Fouling of the air filter can significantly affect the main engine performance and efficiency of the turbocharger, moreover, it could also result in an increase in fuel consumption. Regular maintenance intervals of the air filter should not be neglected and it is recommended to control the amount of fouling more often. Replacing an air filter with a new one or washing an old one is considered as an easy maintenance task and maintenance costs are negligible when compared with potential losses. For compressor wheel faults, it is recommended to perform a maintenance interval according to manufacturer instructions and to continuously monitor scavenge air temperature at the compressor outlet.
- With a high fouling level of the turbine wheel, exhaust gas temperatures are increased and this could lead to damage to the turbine blades. Furthermore, turbine capacity, power and efficiency of the combustion process are reduced. It is necessary to monitor the exhaust temperatures at the turbine inlet and outlet (exhaust ducts). The maintenance interval should be adjusted according to quality of fuel used and the level of contamination.
- The results of the simulation of air cooler faults have indicated that it is a highly important component of the turbocharger system and it should not be neglected in terms of inspection and preventive maintenance. The efficiency of the combustion process and reliability of turbocharging depend on the operating condition of the air cooler.

This method for diagnosing and simulating failures during the operating period is useful to provide analysis of failure causes and to improve the experience in early failure detection. However, turbochargers in low-speed marine engines are complex systems and many unpredictable factors can affect their efficiency. Some problems that are not analyzed in this paper but could also occur during operation are: insufficient ventilation around the engine, lack of lubrication or incorrect lubricating oil, turbocharger shaft misalignment, high ambient air humidity, surging effect, mismatching of the operating engine with the turbocharger system.

More stringent emission regulations and fuel economy are forcing manufacturers to adapt turbocharger performance to new exhaust technologies. With new technologies, there is still much uncertainty in terms of achieving optimal turbocharger efficiency and reliability, such as: how do new alternative fuels impact turbocharger performance? Does two-stage turbocharging have a long-term future? Will turbocharger emission reduction technologies add to maintenance costs? Does slow-steaming reduce turbocharger efficiency?

Although there will be many influences on turbocharger design in the near future, the main priorities will be the reliability of the system, energy efficiency and maintenance costs. For achieving these demands, the methodology presented in this paper is highly useful and practical. Operational efficiency of vessels could be enhanced by this methodology, with real-time scenarios and even simulation of joint bridge-engine operation. The advantage of a marine engine simulator is that failures can be simulated without any consequences for the engine or equipment. Otherwise, this faulty operation of marine engine and turbocharger can be dangerous and impossible under real conditions during voyage. Simulation results and diagnosis of failures can be used as a new educational method for students and useful information for ship owners and engine crew. The presented results can be further used for scientific research in the field of optimization of the main engine and turbochargers. Additionally, the results can be used for the evaluation of the fuel oil consumption and reduction of exhaust gas emissions.

**Author Contributions:** Conceptualization, V.K. and J.O.; methodology, V.K.; research simulation, V.K. and J.O.; validation, V.K., J.O. and L.S.; formal analysis, V.K., J.O. and J.Č.; writing—original draft preparation, V.K.; writing—review and editing, V.K., J.O., J.Č. and L.S. All authors have read and agreed to the published version of the manuscript.

**Funding:** This research received no external funding.

**Conflicts of Interest:** The authors declare no conflict of interest.

## Abbreviations

The following abbreviations are used in the manuscript:

| | |
|---|---|
| $C_{p,c}$ | specific compressor enthalpy |
| $C_{p,t}$ | specific heat of turbine gas |
| $\Delta h_c$ | specific compressor enthalpy |
| $\Delta h_{is,c}$ | isentropic compressor enthalpy |
| $\dot{m}_t$ | turbine gas flow rate |
| P(E) | probability of event |
| $P_t$ | turbine power |
| $T_1$ | compressor inlet temperature |
| $T_2$ | compressor outlet temperature |
| $T_{2,is}$ | isentropic compressor outlet temperature |
| $T_3$ | turbine inlet gas temperature |
| $T_4$ | turbine outlet gas temperature |

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
