# Peer review of "Fault Tree Analysis and Failure Diagnosis of Marine Diesel Engine Turbocharger System"

_jmse, doi:10.3390/jmse8121004_

Round 1

Reviewer 1 Report

My comments which will improve the quality of the paper:

  1. Text line 5.: Please add turbine nozzle ring
  2. Text line 152: Please replace the slow speed by low speed
  3. 4.1. Air filter fault
    In which way the air filter blockage level affects the simulation of the air system? How these fault degrees affect the compressor operation? Is there any direct connection to the compressor performance parameters assumed?
  4. Section 4.3 Compressor wheel fault
    How is the relationship described between degree of fouling and compressor performance?
  5. Section 4.4 Turbine blades fault
    How is the relationship described between degree of fouling and turbine (stage) performance? Especially on the turbine efficiency loss and the turbine flow characteristics modification?

Author Response

Dear Reviewer 1,

Thank you for your time and suggested comments to improve this manuscript.

Please find answers to your comments in attachment.

Best regards,

Authors

Reviewer 2 Report

This manuscript analyzes a marine diesel engine turbocharger system using an engine simulator. Personally, I do not find this research innovative nor useful for the scientific community. For this reason I recommend to reject this manuscript. Other modifications that must be realized:

- A literature review must be included in the Introduction section, providing information about other related works and a critical analysis about them.

- Some type of validation using experimental results must be included in the research since a simulator from Wartsila is applied to a MAN engine.

- Include information about how you obtained Figs. 4-7.

To a lesser extent:

- Line 83, change “method” to “Method”

- Line 90, change “The research paper [11]” to “Adamkiewicz [11]”

- Line 129, change “can’t” to “cannot”

- Line 189, change “Rpm” to “rpm”

- Table 4, change “min.” to “min”

- Line 220, change “chapter 5.5” to “section 5.5”

- Line 271, change “chapter 5.2” to “section 5.2”

- Line 370, change “Table 3.” to “Table 3”

- Line 427, change “Rpm” to “rpm”

- Lines 442, 508, and 520, change “shouldn’t” to “should not”

- Line 448, change “doesn’t have” to “does not have”

Author Response

Dear Reviewer 2,

Thank you for time and comments. The authors improved this manuscript according to your suggestions and answered your comments in details. Please find answers and attached certificate in attachment.

Best regards,

Authors

Round 2

Reviewer 2 Report

The authors have taken into consideration the suggestions. I still think that the manuscript is too poor since it is necessary a rigorous validation procedure for an article to a journal like JMSE. Nevertheless, I recommend it to be published.